# Adsorption Mechanisms and Characteristics of Hg$^{2+}$ Removal by Different Fractions of Biochar

**Xiaoli Guo, Menghong Li \*, Aijv Liu, Man Jiang, Xiaoyin Niu and Xinpeng Liu**

Department of Resource and Environmental Engineering, Shandong University of Technology, Zibo 255000, China; gwmy1027@163.com (X.G.); liuajsdut@gmail.com (A.L.); jiangman@sdut.edu.cn (M.J.); zbnxy@sdut.edu.cn (X.N.); wangbaohua1996@163.com (X.L.)

**\*** Correspondence: zblmh76180@sdut.edu.cn; Tel.: +86-139-5336-9334

**Abstract:** The adsorption mechanisms of mercury ion (Hg$^{2+}$) by different fractions of biochar were studied, providing a theoretical basis and practical value for the use of biochar to remediate mercury contamination in water. Biochar (RC) was prepared using corn straw as the raw material. It was then fractionated, resulting in inorganic carbon (IC), organic carbon (OC), hydroxyl-blocked carbon (BHC), and carboxyl-blocked carbon (BCC). Before and after Hg$^{2+}$ adsorption, the biochar fractions were characterized by several techniques, such as energy-dispersive X-ray spectroscopy (EDS), Fourier-transform infrared spectroscopy (FTIR), and X-ray photoelectron spectroscopy (XPS). Obtained results indicate that the reaction mechanisms of RC for Hg$^{2+}$ removal mainly include electrostatic adsorption, ion exchange, reduction, precipitation, and complexation. The equilibrium adsorption capacity of RC for Hg$^{2+}$ is 75.56 mg/g, and the adsorption contribution rates of IC and OC are approximately 22.4% and 77.6%, respectively. Despite the lower rate, IC shows the largest adsorption capacity, of 92.63 mg/g. This is attributed to all the mechanisms involved in Hg$^{2+}$ adsorption by IC, with ion exchange being the main reaction mechanism (accounting for 39.8%). The main adsorption mechanism of OC is the complexation of carboxyl and hydroxyl groups with Hg$^{2+}$, accounting for 71.6% of the total OC contribution. BHC and BCC adsorb mercury mainly via the reduction–adsorption mechanism, accounting for 54.6% and 54.5%, respectively. Among all the adsorption mechanisms, the complexation reaction of carboxyl and hydroxyl groups with Hg$^{2+}$ is the dominant effect.

**Keywords:** biochar; different fractions; Hg$^{2+}$; characterization; adsorption mechanism

## 1. Introduction

Mercury is a highly toxic, non-essential heavy metal. Even at low concentrations, it poses potential threat to human health [1]. Inorganic mercury is the most common form of mercury found in aquatic ecology. It stably accumulates in aqueous solutions and does not degrade easily. Under natural conditions, it easily reacts with chemicals and microorganisms via a methylation reaction, producing mercury forms with increased toxicity, such as methylmercury or dimethylmercury [2]. According to United States Environmental Protection Agency (EPA) regulations, mercury concentrations in treated wastewater and drinking water must be less than 10 and 2 µg/L, respectively [3]. Therefore, it is of great significance to study mercury removal from water. Treatment methods for mercury removal mainly include chemical precipitation, ion exchange, reduction, coagulation, solvent extraction, and adsorption methods. Among them, adsorption is widely applied due to its advantages of fast action, superior removal efficiency, inexpensive nature of adsorbents, and easy access to raw materials that can be used as adsorbents [4–6]. Biochar is the most commonly used adsorbent material, showing high adsorption efficiency [7]. In previous studies, commonly used raw materials for biochar preparation

included walnut shells [8], flax shive [9], guava skin [10], fruit shell of Terminalia catappa [11], rice husk [12], sugarcane bagasse [13], coconut husk [14], and peanut husk [15], all showing good adsorption results. In general, biochar adsorbs heavy metals mainly via combining metal ions with aromatic alcohols or acids, and carbonates [16]. The main adsorption pathways are through electrostatic effect, reduction, complexation, and cation exchange. Kong et al. [17] showed that the phenolic hydroxyl groups contained in soybean stalk-based biochar had a strong reduction effect, reducing $Hg^{2+}$ to $Hg^0$ during $Hg^{2+}$ adsorption. Das et al. [18] studied $Hg^{2+}$ adsorption by Aspergillus versicolor biomass and suggested complexation as the main mechanism between oxygen-containing functional groups and $Hg^{2+}$. Kílıç et al. [19] observed the release of a large concentration of basic metal ions ($Ca^{2+}$, $Na^+$, and $K^+$) during $Hg^{2+}$ adsorption by activated sludge biomass, indicating that an ion exchange mechanism was involved in the adsorption process. Dong et al. [20] used XPS to analyze the adsorption mechanisms of $Hg^{2+}$ by Brazilian pepper biochars at different pyrolysis temperatures. It was shown that biochar prepared at relatively low temperatures (300 and 450 °C) adsorbs $Hg^{2+}$ mainly through complexation with hydroxyl and carboxyl functional groups, while biomass carbon prepared at 600 °C has a graphite-like structure and Hg–Cπ is formed between $Hg^{2+}$ and the C=C structure. As a heterogeneous material, biochar is mainly composed of inorganic mineral components (ash) and organic carbonaceous components. Current studies on adsorption of pollutants by biochar mainly focus on the overall removal effect of biochar, with research on the adsorption mechanisms of pollutants by different fractions of biochar being very limited. The contribution rate of each fraction of the adsorbent during the adsorption mechanism for a certain pollutant needs to be further studied.

Herein, biochar was prepared using corn stover as the raw material, which was then fractionated to prepare inorganic, organic, hydroxyl-blocked, and carboxyl-blocked carbonaceous fragments. Through physicochemical analysis (element composition, pH value, and isoelectric point $pH_{pzc}$), Boehm titration and surface analysis (FTIR, XPS, and EDS) measurements before and after $Hg^{2+}$ adsorption, the mercury adsorption mechanisms, and structure–property relationships of biochar and its fractions were studied. This research has important practical significance for mercury removal from water by biochar.

## 2. Materials and Methods

### 2.1. Preparation of Biochar Fractions

Raw carbon was synthesized from cornstalk (sourced from Da Xu Village, Zibo, Shandong, China) and air-dried outdoors. It was ground when the moisture content dropped below 10% and was put in a ceramic crucible for pyrolysis and carbonization at 500 °C with a rate of 20 °C/min for 6 h under $N_2$ atmosphere. The biochar removed from the furnace was cooled in a desiccator, weighed, and stored in airtight plastic bags. Raw carbon was referred to as RC. RC was placed in a tube furnace, the temperature of which was raised to 600 °C at a rate of 20 °C/min and kept for 4 h to obtain inorganic carbon (IC). RC was thoroughly mixed for 24 h with an acid mixture (0.3 M HF + 0.1 M HCl), at a solid/liquid ratio of 1:250 (g/mL). The mixture was then washed and dried to obtain organic carbon (OC). Meanwhile, a quantity of 9.0 g of RC was thoroughly mixed with 633 mL of anhydrous methanol and 5.4 mL of 0.1 M HCl to block carboxylic acid groups (–COOH) by methyl esterification of carboxyl groups. After being stirred for 6 h, the mixture was washed and dried to obtained carboxyl-blocked carbon (BCC). Similarly, hydroxyl-blocked carbon (BHC) was prepared by blocking hydroxyl groups (–OH) via thoroughly mixing and stirring 5.0 g of biochar and 100 mL of HCHO for 6 h [21–23]. The yield of each fraction of biochar was calculated.

### 2.2. Physicochemical Properties of Biochar Fractions

Typically, 1.0 g of biochar was weighed and placed in a 100 mL covered conical flask. Meanwhile, 20 mL of deionized water was boiled and let to cool before being added into the flask. Subsequently, the mixture was shaken at 120 r/min and 25 °C for 24 h. It was then taken out and let to sit still for

5 min. The pH of the solution was determined using a Mettler Delta 320 pH Meter (Mettler Toledo, Guangzhou, China) [24].

For a typical test, 50 mL of 0.01 M $KNO_3$ solution was pipetted into a 100 mL covered conical flask. The initial pH value was adjusted to a value within the range of 2–10, using 0.10 M HCl or NaOH solution, and denoted as $pH_i$. Then, 0.1 g biochar was added and the mixture was shaken at 120 r/min and 25 °C, for 48 h. Subsequently, the mixture was filtered and the pH of the filtrate was marked as $pH_f$. In the case where $\Delta pH$ ($\Delta pH = pH_f - pH_i$) is 0, the value of $pH_i$ (or $pH_f$) is the $pH_{pzc}$ of the biochar [25].

A Vario EL cube elemental analyzer (Elementar, Karlsruhe, Germany) was used to determine the C, H, and N content of biochar and its fractions, and the H/C value was calculated.

### 2.3. Determination of Oxygen-Containing Functional Groups on Biochar Surface

Boehm titration [26] was applied to quantitatively determine the oxygen-containing functional groups on the surface of biochar. Specifically, 1.0 g of biochar was placed in a 100 mL covered conical flask, and 25 mL of 0.05 M alkaline solution ($NaHCO_3$, $Na_2CO_3$, and NaOH) and 25 mL of 0.05 M acidic solution (HCl) were added, respectively. The mixture was shaken at 120 r/min and 25 °C for 24 h before being subjected to filtration. Subsequently, back titration was carried out with 0.05 M NaOH and HCl [27] to determine the content of acidic and basic groups on the surface of biochar.

### 2.4. Adsorption Experiment and Characterization before and after $Hg^{2+}$ Adsorption

A $HgCl_2$ 3 mM solution was prepared. Then, 1.0 g of biochar (passed through a 200-mesh sieve) and 200 mL $HgCl_2$ solution were placed into a 250 mL covered conical flask. The mixture was shaken at 120 r/min and 25 °C for 24 h and filtrated upon reaching adsorption equilibrium. The pH of the filtrate was measured, and the concentration of $Hg^{2+}$ was determined via an inductively coupled plasma mass spectrometer (ICP-MS, Santa Clara, CA, USA). On this basis, the $Hg^{2+}$ amount adsorbed was calculated. Meanwhile, the leaching experiments of all biochars were conducted under the same conditions, with samples withdrawn at certain time intervals (3, 5, 10, 20, 30, 60, 90, 120, 150, 180, 240, 300, 420, 540, and 720 min). After filtration, the changes in the concentrations of $K^+$, $Na^+$, and $Hg^{2+}$ in the leaching solution were measured.

The surface functional groups of the samples before and after mercury adsorption were identified by Fourier-transform infrared spectroscopy (FTIR), using a Nicolet 5700 machine (Thermo Nicolet, Waltham, MA, USA). The change in the valence state of $Hg^{2+}$ before and after mercury adsorption was determined by X-ray photoelectron spectroscopy (XPS), using an Axis Ultr DLD machine (Thermo Fisher Scientific, Waltham, MA, USA) and analyzing the experimental data using the software Peak 4.1. (Hampton, MA, USA) The main elemental composition of biochar was detected by an energy-dispersive spectrometer (EDS), using a Quanta 250 (FEI, Hillsboro, OR, USA).

## 3. Results and Discussion

### 3.1. Physicochemical Properties of Biochar and Analysis of Surface Functional Groups

The different biochar fractions examined displayed varied physicochemical properties, as well as difference in the concentration of surface functional groups. According to Lehmann [28] and Shaaban et al. [29], when the pH value of the biochar is greater than $pH_{pzc}$, the biochar surface is negatively charged, which combined with positively charged ions present in the solution, leads to weakened competitiveness of $H^+$. As shown in Table 1, the pH of the biochar was greater than $pH_{pzc}$, so the surfaces of the five biochar fractions are all negatively charged, posing an electrostatic adsorption effect on $Hg^{2+}$. OC has the lowest pH with the highest contents of acidic functional groups such as –OH and –COOH groups. This may be caused due to OC not being cleaned thoroughly after acid washing or due to a replacement of the cations in the functional groups, such as –COOM (M is a metal cation) by $H^+$ during the acid washing process, resulting in the acidity of OC [30,31]. The pH value

of OC did not change significantly after $Hg^{2+}$ adsorption, which may be due to the release of $H^+$ by the complexation reaction between carboxyl/hydroxyl functional groups and $Hg^{2+}$ [32]. $Hg^{2+}$ mainly exists in the anion forms of $HgCl^-$ and $HgClO^-$ in acidic solutions [33]. The carboxyl and hydroxyl functional groups present on OC surface consume $H^+$ by protonation, forming positively charged $-OH^{2+}$ and $-COOH^{2+}$, which combine with $HgCl^-$ and $HgClO^-$ through electrostatic effect to remove $Hg^{2+}$. IC has the highest concentration of strongly basic functional groups, possibly attributed to it being the product of RC pyrolysis and carbonization at 600 °C. Thy et al. [34] showed that after high-temperature pyrolysis, graphitization of biochar occurs via polycondensation of low-temperature aromatization, which removes organic components, resulting in increased alkali metal ions. The pH of IC dropped significantly after $Hg^{2+}$ adsorption, attributed to the fact that $Hg^{2+}$ easily reacts with $OH^-$ to form $Hg_2(OH)_2$ precipitation under alkaline conditions, resulting in decreased concentration of $OH^-$ [35]. Although the concentration of carboxyl and hydroxyl oxygen-containing functional groups in BHC and BCC decreased significantly, the H/C value of BHC and BCC decreased, and the number of lactone groups increased significantly. Dougherty et al. [36] showed that a smaller H/C value represents a lower degree of carbonation and a higher degree of aromatization. They showed that the strength of the cation $-\pi$ effect is mainly determined by the aromaticity degree of biochar surface, i.e., the more abundant the $-\pi$ conjugated aromatic structure, the stronger the electron donating ability of the biochar, and thereby, the more significant the reduction effect. Therefore, BHC and BCC show a strong reduction effect.

**Table 1.** Physicochemical properties and the number of surface functional groups of biochar (mmol/g).

| Biochar | $pH_{pzc}$ | pH | pH after Adsorption | H/C | Carboxyl | Lactone Group | Phenolic Hydroxyl | Acid Functional Groups | Basic Functional Groups |
|---|---|---|---|---|---|---|---|---|---|
| RC | 9.3 | 9.5 | 7.6 | 0.036 | 0.370 | 0.050 | 0.125 | 0.545 | 0.970 |
| IC | 10.4 | 10.9 | 8.6 | 0.314 | 0.105 | 0.210 | 0.080 | 0.395 | 1.390 |
| OC | 3.3 | 3.6 | 3.3 | 0.043 | 0.495 | 0.090 | 0.355 | 0.940 | 0.285 |
| BHC | 8.2 | 8.4 | 6.4 | 0.017 | 0.400 | 0.190 | 0.015 | 0.605 | 0.905 |
| BCC | 8.1 | 8.3 | 6.5 | 0.018 | 0.180 | 0.195 | 0.175 | 0.550 | 0.895 |

RC: raw carbon; IC: inorganic carbon; OC: organic carbon; BHC: hydroxyl-blocked carbon; BCC: carboxyl-blocked carbon.

### 3.2. FTIR Analysis of Biochar Fractions before and after $Hg^{2+}$ Adsorption

The FTIR spectra of different biochar fractions before and after $Hg^{2+}$ adsorption are shown in Figure 1. A significant characteristic peak at 3431 $cm^{-1}$ was observed for all the five biochar fractions. This was assigned to $-OH$ stretching vibration of alcohol hydroxyl groups and phenolic hydroxyl groups formed via intermolecular hydrogen bonding, indicating the large concentration of $-OH$ contained in the biochar fractions [37]. The broad peak around 1578 $cm^{-1}$ was assigned to the stretching vibration of C=O in the lactone group and C=C in mononuclear aromatic hydrocarbons [38], while the stretching vibration of $-COOH$ resulted in the broad peak observed around 1425 $cm^{-1}$ [39]. Therefore, it can be concluded that biochar contains oxygen-containing functional groups, such as $-COOH$, $-OH$, and lactone groups, observations consistent with the results in Table 1.

The absorption peaks weakened after $Hg^{2+}$ adsorption for all the five biochar fractions, with the most significant changes observed for absorption bands at 3431 and 1425 $cm^{-1}$, indicating that the concentration of $-OH$ and $-COOH$ groups were significantly affected by $Hg^{2+}$ adsorption. The absorbance of OC, RC, and BCC at 3431 $cm^{-1}$ decreased by 0.474, 0.327, and 0.185, respectively, after $Hg^{2+}$ adsorption, indicating that the $-OH$ groups possibly reacted with $Hg^{2+}$. This is consistent with the perspective proposed by Herrero et al. [40] that the phenolic hydroxyl groups on the adsorbent surface undergo a complexation reaction with $Hg^{2+}$. At 1425 $cm^{-1}$, the absorbance of OC, RC, and BHC decreased by 0.178, 0.103, and 0.083, respectively, after $Hg^{2+}$ adsorption. Zhang [41] and Goyal [42] et al. also found that the oxygen-containing functional groups (e.g., $-COOH$) on the biochar surface reacted with $Hg^{2+}$, which weakened the vibration signal of $-COOH$ in FTIR.

Before Hg$^{2+}$ adsorption, the concentration of lactone functional groups in BHC and BCC was big (Table 1). After Hg$^{2+}$ adsorption, the absorbance of BHC and BCC at 1578 cm$^{-1}$ decreased by 0.166 and 0.182, respectively, indicating that the C=O in the lactone group and C=C in aromatic hydrocarbons engaged in a reaction with Hg$^{2+}$. A smaller H/C value represents a more significant reduction effect of cation –π, as the –π electron of the benzene ring can reduce Hg$^{2+}$ to Hg$^{+}$ [36,43]. This also proves that BHC and BCC have a strong reduction effect, in accordance with Table 1.

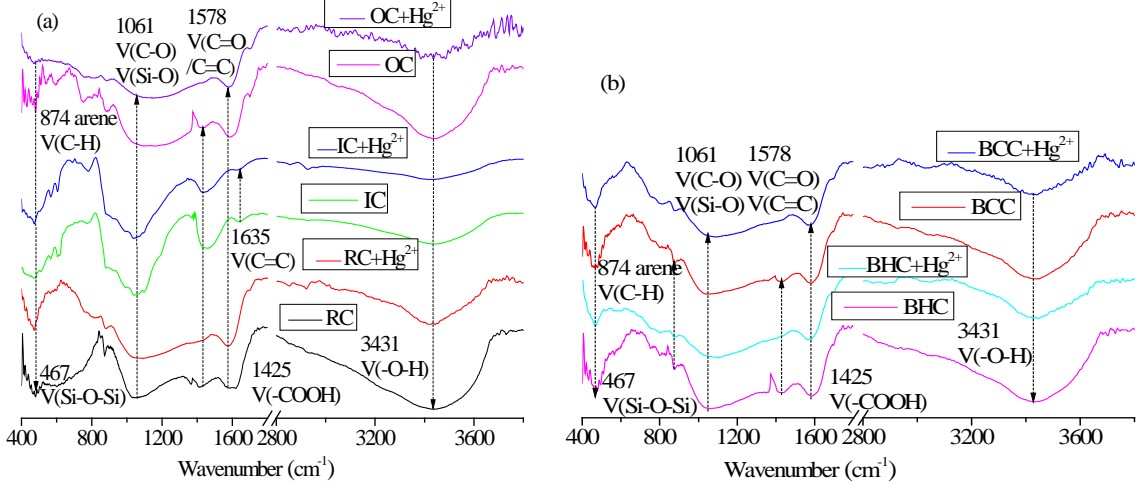

**Figure 1.** FTIR spectra before and after Hg$^{2+}$ adsorption by biochar (**a**): RC, IC, and OC; (**b**): BHC and BCC.

## 3.3. XPS Analysis of Biochar Fractions before and after Hg$^{2+}$ Adsorption

XPS characterization was carried out for biochar fractions before and after mercury adsorption, to analyze the valence changes of C, O, and Hg and to further clarify the adsorption mechanisms of mercury by biochar. Table 2 shows the peak area changes of different forms of C and O in the five biochar fractions before and after mercury adsorption. After mercury adsorption by RC and OC, the contents of carboxyl C (288 eV) decreased by 6.4% and 9.5%, and the contents of carboxyl O (534.6 eV) decreased by 9.6% and 13.0%, respectively. Meanwhile, the contents of hydroxyl O (532.4 eV) decreased by 4.7% and 5.6% in RC and OC, respectively. These results are consistent with the FTIR analysis, indicating that the –COOH and –OH functional groups on the biochar surface complexed with Hg$^{2+}$. The variation of carboxyl and hydroxyl groups on the OC surface is more significant than that on RC surface, attributed to OC being weakly acidic whereas RC is alkaline. Hg$^{2+}$ is more likely to form Hg(OH)$_2$ under alkaline conditions [44], but OC contains more –COOH and –OH functional groups and, therefore, engages more Hg$^{2+}$. The content of hydroxyl O on BCC surface decreased by 7.1%, while the contents of carboxyl C and O on BHC surface dropped by 5.3% and 6.8%, respectively. This indicates that the –COOH groups of BHC and –OH groups of BCC also contribute greatly to mercury removal. After Hg$^{2+}$ adsorption by IC, BHC, and BCC, the areas of C peaks at 284.7 (C=C) and 531.7 eV (C=O) decreased significantly, and the O peak at 531.7 eV decreased by 3.1%, 13.0%, and 15.9%, respectively. These changes indicate that C=C and C=O are also involved in Hg$^{2+}$ removal, by combining with mercury to form Hg–Cπ and remove Hg$^{2+}$ [45].

**Table 2.** Elemental binding energy and mass content of different biochar components before and after $Hg^{2+}$ adsorption.

| Element | BE(eV) [a] | PA(%) [b] of RC | | PA(%) [b] of IC | | PA(%) [b] of OC | | PA(%) [b] of BHC | | PA(%) [b] of BCC | |
|---|---|---|---|---|---|---|---|---|---|---|---|
| | | BS [c] | AS [d] | BS | AS | BS | AS | BS | AS | BS | AS |
| $C_{1s}(C=C)$ | 284.7 | 54.0% | 49.7% | 55.8% | 51.2% | 50.6% | 52.1% | 61.4% | 55.5% | 60.7% | 52.8% |
| $C_{1s}(C–O)$ | 285.6 | 25.7% | 36.7% | 14.0% | 27.1% | 12.9% | 24.1% | 5.7% | 25.4% | 7.5% | 23.6% |
| $C_{1s}(C=O)$ | 287.1 | 8.0% | 7.7% | 22.2% | 18.6% | 16.4% | 15.1% | 22.1% | 13.6% | 27.4% | 19.7% |
| $C_{1s}(–COO)$ | 288.8 | 12.3% | 5.9% | 8.0% | 3.1% | 18.1% | 8.6% | 10.8% | 5.5% | 4.5% | 4.0% |
| $O_{1s}(C=O)$ | 531.7 | 25.5% | 33.2% | 41.2% | 38.1% | 27.4% | 37.7% | 51.5% | 38.5% | 53.4% | 37.5% |
| $O_{1s}(hydroxyl)$ | 532.4 | 23.5% | 18.8% | 15.5% | 11.7% | 25.5% | 16.9% | 7.4% | 7.2% | 21.7% | 14.6% |
| $O_{1s}(C–O)$ | 533.6 | 25.2% | 31.7% | 21.3% | 36.9% | 17.0% | 28.3% | 15.1% | 35.1% | 19.8% | 40.9% |
| $O_{1s}(COO)$ | 534.6 | 25.8% | 16.3% | 21.0% | 13.3% | 30.2% | 17.2% | 26.1% | 19.3% | 5.1% | 5.0% |

[a] BE: binding energy, [b] PA: peak area, [c] BS: before sorption of $Hg^{2+}$, [d] AS: after sorption of $Hg^{2+}$.

The XPS pattern analysis of $Hg_{4f}$ (Figure 2) confirms that a certain amount of $Hg^{2+}$ was adsorbed on biochar. Obvious $Hg_{4f\,7/2}$ and $Hg_{4f\,5/2}$ peaks were observed for RC, IC, and OC, at the binding energies of 101.50 and 101.80 eV, which are two states of spin orbits with a split value of 4.1 eV (distance between two $Hg_{4f}$ peaks), representing the complexes of $(–COO)_2Hg$ and $(–O)_2Hg$ [46,47]. According to the areas of fitted peaks assigned to $(–COO)_2Hg$ and $(–O)_2Hg$, the complexations by IC, RC, and OC account for 39.8%, 68.9%, and 71.6%, respectively, with the corresponding $(–COO)_2Hg$ taking up 18.0%, 51.0%, and 64.6%, respectively. Therefore, it can be concluded that the main adsorption mechanism of RC and OC is a complexation, dominated by –COOH functional groups. Manivannan [48], Wang [49], and Hyland [50] et al. have discovered that the peaks at the binding energies of 102.5 ($Hg_{4f\,7/2}$) and 106.4 eV ($Hg_{4f\,5/2}$) correspond to $Hg^{2+}$; the fitted peak at 103.6 eV is attributed to $Hg_2Cl_2$; and the peak at 99.8 eV is assigned to $Hg^0$. The fitted peak of $Hg^+$ at 103.6 eV is present in the XPS spectra of all the five biochar factions, with the fitted peak of $Hg^0$ at 99.8 eV being observed only for IC (Figure 2). According to FTIR analysis and the significant decrease of the concentration of phenolic hydroxyl groups on biochar (except BHC) after $Hg^{2+}$ adsorption (Figure 1, Table 2), there may be a reduction effect of phenolic hydroxyl groups and –π electrons during mercury adsorption, causing the reduction of $Hg^{2+}$ to $Hg^+$ and $Hg^0$. Such reduction reactions account for 25.2%, 36.4%, and 13.0% in the $Hg^{2+}$ adsorption by RC, OC, and IC, respectively.

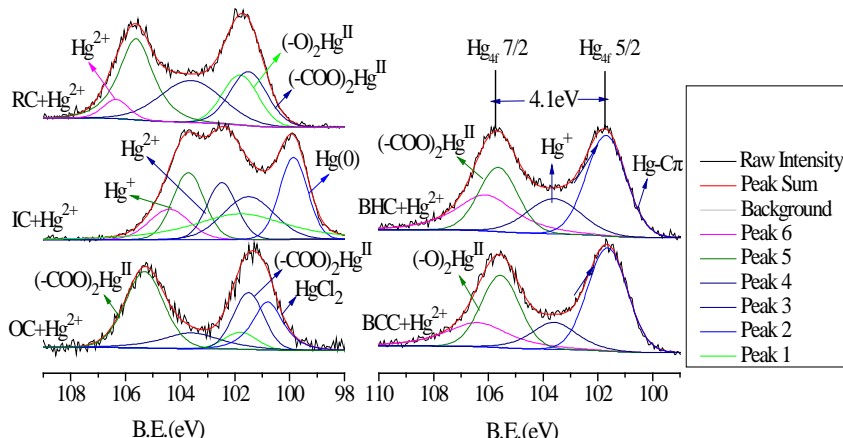

**Figure 2.** XPS spectra of $Hg_{4f}$ after adsorption of $Hg^{2+}$ by biochars.

According to the above analysis, the fitted peaks at 105.6 and 105.9 eV represent $(–COO)_2Hg$ and $(–O)_2Hg$, respectively, indicating that the complexation reaction accounts for 23.3% and 28.3% in the adsorptions by BHC and BCC, respectively. Xu et al. [51] assigned the fitted peak around 101.7 eV to Hg–Cπ, formed from the reduction of $Hg^{2+}$ to $Hg^+$ by the –π bond and the further combination of $Hg^+$ with C=C and C=O. Its peak area accounts for 37.1% and 41.4% in the XPS spectra of BHC and BCC, respectively. According to the areas of fitted peaks at 103.6 eV ($Hg_2Cl_2$) in the XPS spectra of BHC and

BCC, reduction accounts for 17.5% and 13.1%, respectively. Thus, the overall reduction effect of BHC and BCC contribute 54.6% and 54.5%, respectively. Therefore, the main adsorption mechanism of BHC and BCC is by reduction.

### 3.4. EDS Analysis of Biochar Fractions before and after $Hg^{2+}$ Adsorption and Leaching Experiments

According to the EDS spectra of the five biochar fractions before and after $Hg^{2+}$ adsorption and leaching experiments of all biochar, IC contains the most basic cations. Therefore, only IC analysis results were displayed. As shown in Figure 3a, after $Hg^{2+}$ adsorption by IC, the concentration of metal cations, such as $K^+$, $Na^+$, $Mg^{2+}$, and $Al^{2+}$, on the biochar decreased, accompanied by the appearance of a relatively obvious mercury peak, indicating that there may be an ion exchange mechanism during the adsorption process. Among the various cations, the most significant reduction was observed for $K^+$, followed by $Na^+$, implying that $Hg^{2+}$ is more prone to ion exchange with $K^+$ and $Na^+$. As suggested by Figure 3b, the $K^+$ and $Na^+$ concentrations gradually increased with time, and the $Hg^{2+}$ concentration simultaneously decreased, until the adsorption equilibrium. The adsorption contribution rates of $K^+$ and $Na^+$ (30.6% and 10.3%, respectively) were calculated, and based on the concentration of $K^+$, $Na^+$, and $Hg^{2+}$ at the adsorption equilibrium, the adsorption contribution rate of ion exchange was 40.9%. According to studies carried out by Kílíç [19] and Carro et al. [52], $K^+$ and $Na^+$ are more prone to ion exchange with $Hg^+$. The contents of $Hg^+$ and $Hg^{2+}$ determined by XPS analysis are 17.3% and 23.9%, respectively. Hence, in the total ion exchange of 40.9%, $Hg^+$ and $Hg^{2+}$ take up 17.3% and 23.6%, respectively. Associating with XPS analysis, the complexation and reduction reactions during $Hg^{2+}$ adsorption by IC account for 39.8% and 36.4%, respectively. Therefore, the main mechanism for $Hg^{2+}$ adsorption by IC is ion exchange. In addition, the adsorption contribution rates of ion exchange in the removal of $Hg^{2+}$ (40.9%) by IC is higher in comparison with other adsorbents such as 31.7% by activated sludge biomass [19] and 35.0% by algal biomass [52].

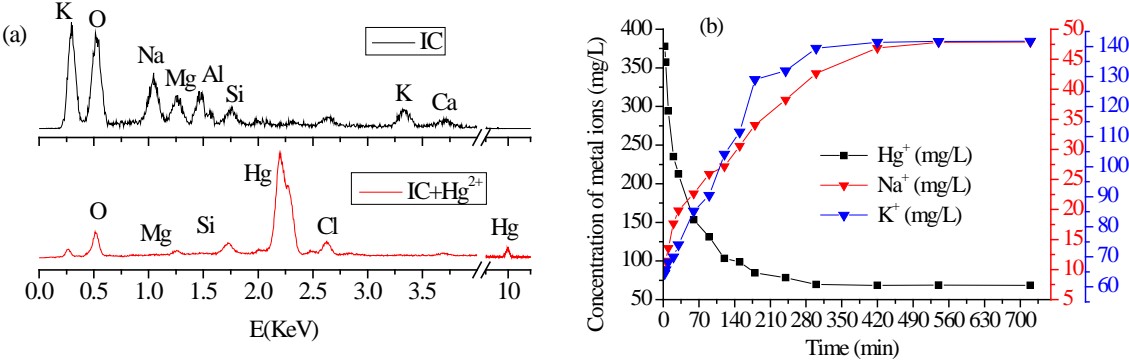

**Figure 3.** (**a**) EDS analysis before and after adsorption of $Hg^{2+}$ by IC; (**b**) The concentration change of $K^+$, $Na^+$, and $Hg^{2+}$ with oscillation time after adsorption of $Hg^{2+}$ by IC.

### 3.5. Adsorption Mechanisms

All five biochar fractions have negatively charged surfaces and can adsorb positively charged $Hg^{2+}$ via, for example, electrostatic effects (Table 1). According to the above analysis, the main adsorption mechanisms and contribution rates of $Hg^{2+}$ adsorption by RC and its fractions were determined. Based on Table 3, the equilibrium adsorption amount of $Hg^{2+}$ by RC is 75.56 mg/g. As calculated based on the mass fractions of IC and OC, the theoretical adsorption capacity of RC is 75.52 mg/g, which is very close to the measured value of 75.56 mg/g, indicating that the adsorption contribution rates of inorganic and organic components in RC account for approximately 22.4% and 77.6%, respectively. The main adsorption mechanism of RC and OC in the $Hg^{2+}$ adsorption process is based on a complexation reaction, where oxygen-containing functional groups, such as –COOH and –OH groups, react with $Hg^{2+}$ to form (–COO)$_2$Hg and (–O)$_2$Hg complexes. The complexation effects of RC and OC contribute

68.9% and 71.6%, respectively. In the complexation reaction of RC, the contribution rates of the adsorption by –COOH and –OH groups are 51.0% and 17.9%, respectively. The reaction formulae are as follows [17]:

$$2 \text{ (-COOH)} + \text{Hg}^{2+} \rightarrow \text{(-COO)}_2\text{-Hg} + 2\text{H}^+, \tag{1}$$

$$2 \text{ (-OH)} + \text{Hg}^{2+} \rightarrow \text{(-O)}_2\text{-Hg} + 2\text{H}^+. \tag{2}$$

IC has the maximum equilibrium capacity for $\text{Hg}^{2+}$ adsorption of 92.63 mg/g. This may be due to the reduction of $\text{Hg}^{2+}$ to $\text{Hg}^+$ and $\text{Hg}^0$ by $-\pi$ electrons and phenolic hydroxyl groups (36.4%), the complexation reaction between carboxyl/hydroxyl functional groups and $\text{Hg}^{2+}$ (39.8%), and the ion exchange of $\text{Hg}^{2+}$ and $\text{Hg}^+$ with $\text{K}^+$ and $\text{Na}^+$ (40.9%) occurring simultaneously. The main adsorption mechanism of IC is ion exchange, during which the ion exchange of $\text{Hg}^+$ and $\text{Hg}^{2+}$ contributes 17.3% and 23.6%, respectively.

As shown in Table 3, the equilibrium capacities for $\text{Hg}^{2+}$ adsorption by BHC and BCC are significantly lower compared to IC, being 66.30 and 61.13 mg/g, respectively. In addition, the calculated theoretical adsorption capacities of BHC and BCC, based on their mass fractions, are 65.96 and 61.43 mg/g, respectively, which are close to the measured values. The $-\pi$ electrons and phenolic hydroxyl groups on the surfaces of BHC and BCC reduce $\text{Hg}^{2+}$ to $\text{Hg}^+$, and total reduction ratios take up 54.6% and 54.5%, respectively. According to the adsorption capacities of BHC and BCC, the adsorption contribution rates of the –COOH groups on BHC surface and the –OH functional groups on BCC surface are 20.4% and 12.3%, respectively. As for RC, –COOH and –OH groups contribute 51.0% and 17.9% respectively, to the total adsorption. The reason for this difference may be attributed to the presence of some –COOH and –OH functional groups remaining after the blocking of the biochar functional groups (Table 1).

**Table 3.** Contribution rate (%) and adsorption quantity of $\text{Hg}^{2+}$ (mg/g) at equilibrium by biochars.

| Biochars | Mass Percentage (%) | Adsorption Quantity of Hg²⁺ (mg/g) | Adsorption Mechanism | Adsorption Contribution Rate (%) |
|---|---|---|---|---|
| RC | 100.0 | 75.56 | Complexation with –COOH and –OH | 68.9 |
| | | | Reduction reaction | 25.2 |
| IC | 22.4 | 92.63 | Complexation with –COOH and –OH | 39.8 |
| | | | Reduction reaction | 36.4 |
| | | | Cation exchange | 41.2 |
| OC | 77.6 | 70.57 | Complexation with –COOH and –OH | 71.6 |
| | | | Reduction reaction | 13.0 |
| BHC | 87.3 | 66.30 | Complexation with –COOH | 23.3 |
| | | | Reduction reaction | 54.6 |
| BCC | 81.3 | 61.13 | Complexation with –OH | 28.3 |
| | | | Reduction reaction | 54.5 |

The adsorption mechanisms of $\text{Hg}^{2+}$ removal by RC include electrostatic adsorption, ion exchange, reduction, precipitation, and complexation. Among all adsorption mechanisms, the complexation reaction of –COOH/–OH groups with $\text{Hg}^{2+}$ plays the dominating roles, and the adsorption contribution rate of –COOH functional groups is greater than that of –OH functional groups.

The adsorption capacity of $\text{Hg}^{2+}$ by corn-straw-based biochar (75.56 mg/g) was found to be higher than some values reported in literature, such as 35.71 mg/g by sugarcane bagasse [13], 22.82 mg/g by peanut husk biochar [15], 19.30 mg/g by activated sludge biomass [19], and 24.20 mg/g by Brazilian pepper biochars [20]. It is, however, similar to the value determined for Aspergillus versicolor biomass (75.60 mg/g) [18], and slightly lower than the adsorption capacities reported for flax shive sorbent (89.50 mg/g) [9] and fruit shell of Terminalia catappa (82.93 mg/g) [11]. Moreover, the irreversible complexation reaction of –COOH/–OH groups with $\text{Hg}^{2+}$ is the main adsorption

mechanism, which reduces the secondary pollution to aqueous solution even depositing it to the water bottom. Therefore, it is helpful to provide theoretical basis and data support for the special adsorption of functional adsorbents and has practical significance for the treatment of mercury pollution in water. In addition, by studying the adsorption mechanisms of biochar different fractions for $Hg^{2+}$, the contribution rate of each fraction of the adsorbent to $Hg^{2+}$ was calculated, providing a certain theoretical basis for the research on the removal of Hg from biochar.

## 4. Conclusions

Except for OC, which is weakly acidic, the other four biochar fractions are basic (IC has the strongest basicity), and their pH values are significantly lower after $Hg^{2+}$ adsorption. In addition, the pH values of all five biochar fractions are greater than their corresponding $pH_{pzc}$ values. The surfaces of the biochar fractions are negatively charged and adsorb positively charged $Hg^{2+}$.

The equilibrium adsorption capacity of RC for $Hg^{2+}$ is 75.56 mg/g, with IC and OC contributing 22.4% and 77.6%, respectively. IC has the largest equilibrium adsorption capacity of 92.63 mg/g, which may be due to all the adsorption mechanisms involved in the adsorption process. Moreover, the equilibrium adsorption capacities of BHC and BCC are significantly lower, being 66.30 and 61.13 mg/g, respectively.

There are five reaction mechanisms identified for $Hg^{2+}$ removal by biochar: electrostatic adsorption, cation exchange, precipitation, reduction effect of $-\pi$ and phenolic hydroxyl groups, and complexation by $-OH$ and $-COOH$ groups. The main adsorption mechanism of RC and OC is the complexation of oxygen-containing functional groups with $Hg^{2+}$, accounting for 68.9% and 71.6%, respectively. The contribution rates of $-COOH$ and $-OH$ groups in RC complexation reaction are 51.0% and 17.9%, respectively, while those in OC complexation reaction are 64.6% and 7.0%, respectively. Among the five examined fractions, IC plays a major role in mercury adsorption by RC. All adsorption mechanisms identified are involved in the adsorption process by IC, with ion exchange being the main one, accounting for 39.8%. The reduction effect of phenolic hydroxyl groups and $-\pi$ electrons is the main mechanism for $Hg^{2+}$ removal by BHC and BCC, taking up 54.6% and 54.5% of the total adsorption for each fragment, respectively. In addition, the adsorption contribution rates of $-COOH$ groups on BHC surface and $-OH$ functional groups on BCC surface are 20.4% and 12.3%, respectively. Among all the adsorption mechanisms, the complexation reaction of $-COOH/-OH$ groups with $Hg^{2+}$ plays the dominating roles, and the adsorption contribution rate of $-COOH$ functional groups is greater than that of $-OH$ functional groups. These conclusions play a guiding role in the remediation mechanism of heavy metals by biochar in water. The prepared RC could be a promising low-cost adsorbent for the treatment of $Hg^{2+}$-contaminated water.

**Author Contributions:** Methodology, M.L.; data curation and writing—original draft preparation, X.G.; supervision, A.L.; funding acquisition, M.J. and X.N.; formal analysis, X.L. All authors have read and agreed to the published version of the manuscript.

**Funding:** This research was funded by the National Natural Science Foundation of China (41771348, 41877122, 41671322, 41703099); Doctor of Natural Science Foundation of Shandong province, China (ZR2018BEE015, ZR2019BB045); Youth Science Foundation of the National Natural Science Foundation of China (51804188).

**Acknowledgments:** The authors are also thankful to the analytical testing center of Shandong University of Technology.

**Conflicts of Interest:** The authors declare no conflict of interest.

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
