# Peer review of "Adsorption Mechanisms and Characteristics of Hg2+ Removal by Different Fractions of Biochar"

_water, doi:10.3390/w12082105_

Round 1

Reviewer 1 Report

This is a nice and thorough contribution to an important area. However, there are several papers that are quite similar. These papers are cited but I think that the results here should be compared explicitly to the other chars in terms of Hg adsorbed per gram of sorbent. For example, walnut shell biochar adsorbed 152 mg/g and flax shives were claimed to adsorb more than 1 g Hg/g char.
On about line 200 in the XPS section, as good as the data are, reporting percent areas to two decimal places is more than can be justified. Especially since XPS curves tend to be fitted prior to integration. Drop at least one digit.
Figure 2. XPS binding energy is usually plotted high energy to low.
I don’t think the xrd diffractograms contribute enough free-standing information to justify keeping it in the manuscript. The EDS data suffice. Although, I don’t know what this INCA machine (Oxford Instruments) is. Is this X-ray fluorescence or an attachment to an SEM? This needs to be clarified.
Table 1 includes the H/C ratio with units of %. I don’t think this is what you meant-it’s just a ratio.
Table 3. The adsorption rate data are not a rate without a time period.
Section 2.1. The experimental details are lacking. It is not clear how the RC is made from the corn cob. Was the cob ground to a sieve size and then torrified? You should also explain that the BCC was prepared through the methyl esterification of the carboxy groups. This chemistry won’t be obvious to all readers.
Finally, comments about what a plan might be for the mercury once it is adsorbed. Can it be recovered? Fated for a landfill? Where do we go from here?

Author Response

Dear reviewer:

Reviewer 2 Report

See attached file.

Author Response

Dear reviewer:

Round 2

Reviewer 1 Report

This paper is a nice contribution.

Author Response

Dear Reviewer and Editor:

Thank you for your letter and for the reviewer’s comments concerning our manuscript entitled “Adsorption mechanisms and characteristics of Hg2+ removal by different fractions of biochar” (ID: water-864141). Those comments are all valuable and very helpful for revising and improving our paper, as well as the important guiding significance to our researches.

Reviewer 2 Report

The manuscript was significantly improved. My only regret is that the authors decided to compare their results regarding sorption capacity only with the weaker results (Lines 300-304). They should also present better results published in other articles and comment on them. I suggest correcting this blunder.

Author Response

Dear Reviewer and Editor:

   Thank you for your letter and for the reviewer’s comments concerning our manuscript entitled “Adsorption mechanisms and characteristics of Hg2+ removal by different fractions of biochar” (ID: water-864141). Those comments are all valuable and very helpful for revising and improving our paper, as well as the important guiding significance to our researches. We have studied comments carefully and have made correction which we hope meet with approval. Revised portion are marked in red in the manuscript. The main corrections and the responds to the reviewer’s comments are as flowing:

Response to revieewer: Lines 300-304 (After modification):"The adsorption capacity of Hg2+ by corn straw-based biochar (75.56 mg/g) is found to be much higher in comparison with most other adsorbents such as 89.50 mg/g by flax shive sorbent [9], 82.93 mg/g by fruit shell of Terminalia catappa [11], 35.71 mg/g by sugarcane bagasse [13], 22.82 mg/g by peanut husk biochar [15], 75.60 mg/g by Aspergillus versicolor biomass [18], 19.30 mg/g by activated sludge biomass [19], and 24.20 mg/g by Brazilian pepper biochars [20]".